# The Status of Wild Grapevine (*Vitis vinifera* L. subsp. *sylvestris* (C.C. Gmel.) Hegi) Populations in Georgia (South Caucasus)

**DOI:** 10.3390/plants14020232

**Published:** 2025-01-15

**Authors:** Gabriele Cola, Gabriella De Lorenzis, Osvaldo Failla, Nikoloz Kvaliashvili, Shengeli Kikilashvili, Maia Kikvadze, Londa Mamasakhlisashvili, Irma Mdinaradze, Ramaz Chipashvili, David Maghradze

**Affiliations:** 1Department of Agricultural and Environmental Sciences, University of Milan, 20133 Milano, Italy; gabriella.delorenzis@unimi.it (G.D.L.); osvaldo.failla@unimi.it (O.F.); 2National Wine Agency of Georgia, Tbilisi 0159, Georgia; nikoloz.kvaliashvili@wine.gov.ge (N.K.); irma.mdinaradze@wine.gov.ge (I.M.); 3Faculty of Viticulture-Winemaking, Caucasus International University (CIU), Tbilisi 0141, Georgia; shengeli.kikilashvili@ciu.edu.ge (S.K.); maia.kikvadze@ciu.edu.ge (M.K.); 4Scientific-Research Center of Agriculture, Tbilisi 0159, Georgia; info@srca.gov.ge; 5Kakha Bendukidze University Campus, Agricultural University of Georgia, Tbilisi 0159, Georgia; r.chipashvili@agruni.edu.ge

**Keywords:** wild grapevine, South Caucasus, Jighaura collection, ampelography, phenology, geography

## Abstract

Repeated expeditions across various regions of Georgia in the early 2000s led to the identification of 434 wild grapevine individuals (*Vitis vinifera* L. subsp. *sylvestris* (C.C. Gmel.) Hegi) across 127 different sites, with 45% of these sites containing only a single vine and only 7% more than 9 vines. A total of 70 accessions were propagated in a germplasm collection, 41 of them were descripted from the ampelographic point of view and 32 from the phenological one. The geographical and ecological analysis confirmed that wild grapevines primarily grow in humid environments with warm and fully humid climates, often near rivers. They favor deep, fertile, and evolved soils, mainly alluvial and cinnamonic types (80%), with a marginal presence on strongly eroded soils. Their main natural vegetations are forests and open woodlands, with some individuals in the Southeast found in steppes. The altitudinal range spans from 0 to 1200 m, with 80% of vines distributed between 400 and 900 m. The phenological analysis revealed significant differences among the accessions but no difference among populations, with only a slight variation in bud-break timing, indicating a high level of synchronicity overall. Flowering timing proved to be the most uniform stage, suggesting minimal environmental pressure on genetic adaptation. The mature leaf morphology exhibited significant polymorphism, though leaves were generally three- or five-lobed, weak-wrinkling, and -blistering, with a low density of hairs. Bunch and berry morphology were more uniform. Bunches were consistently very small, cylindrical, and never dense or winged. Berries were also very small, mostly globular, always blue-black in color, and non-aromatic. A striking feature was the frequency of red flesh coloration, which ranged from weak to strong, with uncolored flesh being rare. The Georgian population of wild grapevines was found to be fragmented, often consisting of scattered single individuals or small groups. Therefore, we believe it is urgent for Georgia to implement specific protection measures to preserve this vital genetic resource.

## 1. Introduction

The wild *Vitis vinifera* L. species, typically referred to as *Vitis vinifera* L. subsp. *sylvestris* (C.C. Gmel.) Hegi to distinguish it from domestic forms (generally called *Vitis vinifera* subsp. *sativa* (DC.) Hegi), belongs to the Mediterranean, sub-Mediterranean, Pontic, Caucasian, and Caspian floristic regions, with an extension toward Central Asia and northern Iran [1,2,3].

Over recent decades, numerous studies have provided information, with varying levels of detail, on the status and characteristics of wild *Vitis vinifera* L. populations (hereafter referred to as wild grapevines). These studies confirm the sporadic distribution of the species across its range. Despite a few attempts to give a general overview of the species’ status [4,5,6,7], a significant number of local and regional reports have been published. From the western to the eastern part of its range, the following contributions can be cited: Benito et al. [8], Cunha et al. [9], De Andrés et al. [10], Iriarte-Chiapusso et al. [11], Ocete et al. [12] for the Iberian Peninsula; Zinelabidine et al. [13] for Morocco; Selmi et al. [14] for Tunisia; Arnold et al. [15] and Lacombe et al. [16] for France; Biagini et al. [17] and Schneider et al. [18] for Italy; Arnold et al. [19] for Germany; Regner et al. [20] for Austria; Zdunić et al. [21,22], Perko et al. [23], Kullaj et al. [24], and Susaj et al. [25] for the eastern Adriatic region; Bodor et al. [26], Bartha et al. [27] (2012), and Jahnke et al. [28] for Hungary; Popescu et al. [29] for Romania; Dzhambazova et al. [30] for Bulgaria; Ergül et al. [31] and Karataş et al. [32,33] for Turkey; Amanov [34], Ocete Rubio et al. [35], Maghradze et al. [36,37], and Kikvadze et al. [38] for the South Caucasus; Rahimi et al. [39] for Israel/Palestine; and Naqinezhadet al. [40] for northern Iran. Unfortunately, there is no recent information regarding the Central Asian wild grapevine populations in European languages (some old references are available in Russian), despite their potential role in grapevine domestication [41].

Wild *Vitis vinifera* is a dioecious and deciduous woody liana that primarily grows in temperate biomes. In fertile floodplains and riparian forests, it assumes Raunkiaer’s phanerophyte life form [4]. However, in limiting soil conditions, it can modify its habitus to become a low-growing shrub, adopting Raunkiaer’s chamaephyte life form, like, for example, in Italian Central Apennine [42] or close to the springs of the Tigris River [43].

According to recent surveys, wild grapevines must be considered an endangered species throughout all their distribution ranges. This is due to the loss and fragmentation of their main natural vegetations (riverbanks and flooded forests), in addition to the management practices of forests and field edges, and the biotic pressure exerted by fungal diseases and pests introduced to Eurasia in the latter half of the 19th century from America. These include powdery mildew *Uncinula necator* (Schwein., Burrill), downy mildew *Plasmopara viticola* (Berk. and M.A. Curtis, Berl. and De Toni), and the phytophagous insect *Daktulosphaira vitifoliae* (Fitch). Additionally, there is potential competition from non-vinifera *Vitis* species [44,45] and genetic introgression from domesticated *Vitis vinifera* forms [46,47].

The Georgian wild grapevine populations are significant for their potential contribution as ancestors of domesticated grapevine varieties. Indeed, genomic and archeological research accredits the South Caucasus as a primary center for *Vitis vinifera* domestication [48,49,50,51,52,53].

The first extensive description of Georgian wild grapevine populations dates to the mid-19th century [54] when their spread was more extensive than it is today. The decline during the 20th century has been attributed to the invasion of harmful American pests and diseases, as well as the human impact on their natural habitat [35]. A. Negrul [55] provided information about wild grapevines of Colchis (western Georgia) during the 1950s, and Maksime and Revaz Ramishvili began systematic research on the distribution and characteristics of wild *Vitis vinifera* in Georgia. By the late 1980s [56,57], they published a distribution map of wild grapevine populations (Figure 1), highlighting eight main distribution areas across the country. According to this map, the distribution area is wider in the eastern part than in the western one, suggesting that high humidity and precipitation and warmer climates stimulated the aggressive character of the fungal pathogens in western Georgia, with a negative impact on wild grape populations.

In recent years, Ekhvaia and Akhalkatsi [58] conducted an ampelographic and ampelometric study of leaf characteristics and flower morphometrics in seven populations of wild grapevine (one hundred and ninety individuals) located in four river basins across three geographic regions of Georgia (western, eastern, and southern). Their results confirmed the dioecious nature of the species, with separate male and functional female plants, and an average male-to-female ratio of 1:2. An uncommon trait highlighted in this survey was the white skin color of berries found in two individuals in a southern Georgian population. Leaf pubescence varied significantly among individuals within the populations, ranging from glabrous leaves to arachnoid and hirsute pubescence, with no occurrences of tomentose pubescent leaves. Leaf morphology showed significant discriminant differences among the different geographic groups: Western Georgian populations had the smallest leaves, usually with three lobes; southern Georgian populations had leaves with both three and five lobes; and eastern Georgian populations had leaves with seven lobes. An interesting polymorphism was observed in flower morphology: western Georgian populations had the longest nectaries, eastern Georgian populations had the shortest, and southern Georgian male flowers had medium-length nectaries. In conclusion, the results revealed significant differences among populations located in different geographic regions of Georgia.

From the genetic point of view, a large molecular survey based on 20 nuclear SSR markers, that involved 1378 wild and cultivated grapevines collected around the Mediterranean basin and from the South Caucasus and Central Asia, highlighted the genetic identity and diversity of the wild grapevines’ population from the other wild populations and, at the same time, closeness with the domestic compartment [51]; similar conclusions were drawn by Ekhvaia et al. [59] by analyzing a smaller sample of wild and domestic Georgian grapevines’ accessions.

In 2003, new survey activities on Georgian wild grapevine populations began as part of various national and international research and germplasm conservation projects as testified by several scientific publications [35,36,38,60,61].

The present paper focuses on the distribution range and ecological, morphological, and phenological characteristics of wild *Vitis vinifera* L. populations in Georgia. In particular, the aims of the work were: (i) to give an updated figure of the current distribution range of the wild grapevine populations in Georgia with respect to previous surveys, their consistency, and the main ecological descriptors of their habitats; and (ii) to organize an ex situ germplasm collection to preserve this supposed endangered species, to characterize it in the same environmental conditions, and to evaluate its role in grapevine breeding for resistance to biotic and abiotic stress.

## 2. Results

### 2.1. Ampelography

The main statistics of the OIV ampelographic harmonized descriptors [62] (Table A1) are reported in Table 1 and referred to forty-one accessions from the Jighaura germplasm collection (GEO038). The different traits are characterized by different polymorphisms.

As expected, the two species-specific traits that characterize *Vitis vinifera* in comparison to non-vinifera *Vitis* species were always confirmed and uniform within the sample of accessions involved in the ampelographic records. More precisely, the “aperture of the young shoot tip” (OIV code 001) was constantly “5 = fully open”, and the “number of consecutive tendrils on the shoot” (OIV code 016) was uniformly “1 = two or less”.

In relation to the young shoot apex traits, a certain degree of polymorphism was detected in terms of the “intensity of anthocyanin coloration on prostrate hairs of the shoot tip” (OIV code 003), with a prevalence of “none or very low” ratings with respect to “low” and “medium” or “high” coloration, while a high polymorphism was distinguished for the “density of prostrate hairs on the shoot tip” (OIV code 004), of which the level of expression ranged from “none or very low” to “high” with the three intermediate levels mostly expressed.

The “shoot attitude” (OIV code 006) was more frequently “semierect” or “horizontal” and no accession was classified as having a “semi dropping” or “dropping” growth habit.

The “color of the ventral and dorsal side of the internodes” (OIV codes 007 and 008) was mainly “red and green” and “green”, respectively.

A very high level of polymorphism has been detected for what concerns the younger leaf blade, in terms of “color of upper side” (OIV code 051) and “density of prostrate hairs between main veins on lower side” (OIV code 053). The high frequency of the “bronze” and “cupper-reddish” colors of the upper side of the leaflets does not match the expected level of expression proposed by Zdunić et al. [21].

The “number of inflorescences per shoot” (OIV code 153) mainly ranged from 1.1 to 3.0; the “fertility of basal buds” (OIV code 155) generally was “very high”.

The mature leaf ampelographic traits showed a large polymorphism. Generally, the leaves were wedge-shaped or pentagonal (OIV code 067) and three- or five-lobed (OIV code 068), with weak goffering (OIV code 072) and blistering (OIV code 075), both sides with convex teeth (OIV code 076), and an open sinus angle, U- or V-shaped (OIV codes 079 and 080), with low density of prostrate hairs between the main veins, and none or low density of erect hairs on the main veins (OIV code 084 and 87).

Bunch and berry morphology were quite uniform. Bunches always had very low weight, were very short or short in length (OIV code 203), cylindric in shape (OIV code 208), never dense (OIV code 204), and never winged (OIV code 209). Berries had very low weight (OIV code 503), were very short or short in length (OIV code 220), mainly globular in shape (OIV code 223), always blue-black pigmented (OIV code 225), and never with a particular flavor (OIV code 236). An unexpected and interesting variable trait was the intensity of flesh anthocyanin coloration (OIV code 231), which was rarely absent or very low and generally ranged from weak to strong.

The frequency distribution of the polymorphic traits is reported in Figure A1.

### 2.2. Phenology

The phenological development of 32 accessions from the Jighaura collection for 2019, 2020, and 2021 is presented in Figure 2, and the main statistical properties for BBCH [63] stages 9, 65, 75, and 85 are presented in Table 2. In all the seasons, the variability of phenological timing was particularly limited. During the three seasons, the bud-break stage BBCH 9 showed time spans of 13, 13, and 6 days, respectively, with a standard deviation of about 3 days. Full flowering (BBCH 65) was the less variable stage with 6-, 5-, and 10-day time spans in 2019, 2020, and 2021, with very low standard deviations.

To understand if some kind of genetic diversification in relation to the different climatic conditions happened among the different wild grapevine populations in Georgia, the relationship between phenological timing (related to BBCH stages 9, 65, 75, and 85), elevation, and longitude was investigated by means of multiple linear regression.

BBCH 9 bud break was the only stage to show a significant difference (Figure 3, Table 3).

More in detail, populations located at higher elevations (that happens to be in the eastern part of Georgia) generally show an average advance in the timing of bud break, when compared to other populations. An explanation of this advance could be explained by a lower request for thermal resources during the eco-dormancy phase for grapevines growing at higher altitudes and used to lower spring temperatures.

The absence of significant relationships for most of the phenological stages considered highlights the absence of a genetic variation in phenological timing induced by the different environmental conditions on the Georgian wild grapevines.

### 2.3. Consistency and Ecology of the Georgian Wild Grapevine Populations

Based on our investigations, the Georgian wild grapevine population was found to be extremely reduced and fragmented across sites hosting single individuals or very small groups of plants: 45% of the sites contained a single individual, 18% contained two, 30% hosted between three and nine individuals, and only 7% hosted more than nine individuals (Figure 4).

The current wild grapevine distribution dataset, composed of 434 records of wild plants, was related to botanical, environmental, geographical, and ecological features. Figure 5, Figure 6, Figure 7, Figure 8, Figure 9 and Figure 10 present maps of the locations of wild grapevine populations, together with the main geographical features of Georgia. Each map is accompanied by a distribution chart showing the populations in relation to the geographical features being investigated.

For 24.2% of the surveyed vines (105 out of 434), it was possible to determine or infer the sex. A male-to-female ratio of less than 1 (approximately 0.84) was observed (Table A2 and Table A3).

As shown in Figure 5, most of the wild grapevine populations fall within the Köppen climate types of Dfb-continental/fully humid/warm summer (45.2%) and Cfa–warm temperate/fully humid/hot summer (47.5%).

The fully humid warm and hot climates are consistent with the grapevines’ natural vegetation. Furthermore, it is interesting to note that the two climate types are representative of 17% of Europe’s wine-growing area [64].

With reference to regional distribution (Figure 6), most of the sites were found in eastern Georgia, with 82.9% of the total number covered by Kakheti, Kvemo Kartli, Shida Kartli, and Mtskheta-Mtianeti, accounting for 37.3%, 17.3%, 9.9%, and 18.4%, respectively. In the central western part of Georgia, the highest number of sites was found in Racha–Lechkhumi and Samtskhe–Javakheti, both accounting for 4.8%. Regarding the comparison with the Ramishvili (2001) survey map, it is important to highlight that Abkhazeti was not covered by the survey, due to recent political issues.

The altitudinal distribution of wild grapevines (Figure 7) shows a peak at 600–700 m a.s.l. (23%), with 76% of the whole population in the 400–900 m range.

Considering soil type (Figure 8, Table A4), about 70% of the sites fall into four soil types: 44 are alluvial calcareous (calcaric fluvisoils), 32 are cinnamonic calcareus, 30 are cinnamonic leached, and 31 are cinnamonic, representing 34.6%, 18.7%, 8.8%, and 7.6%.

Considering altitudinal distribution and soil type together, it is interesting to note that wild grapes below 400 m of altitude were found mainly on alluvial soils in the eastern regions of Kakheti, Kvemo Kartli, Shida Kartli, and Mtskheta-Mtianeti. On the other hand, wild vines above 800 m were primarily found in Kakheti and Samtskhe–Javakheti on cinnamonic soils and secondly on alluvial soils in Mtskheta-Mtianeti. Most of the vines were found in the 400–800 m range, with a greater variability of soil types and the prevalence of cinnamonic soils.

In total, 44.2% of the sites were detected in the Mtkvari water catchment basin (Figure 9), 21.4% in the Alazani basin, and 15.9% in the Iori basin. It is interesting to observe that 29% of populations were found within 100 m distance from the river, about 50% within 300 m, and about 77.6% within 500 m.

Regarding vegetation formation (Figure 10, Table A5), 63.8% of the sites falls into F macro-class mesophytic deciduous broad-leaved and mixed coniferous–broad-leaved forests (F163, F164, F165, F169, and F170) and can be found across all the Georgian regions. Vegetation formation type F170 alone covers 44% of the total number of vines and can be found in all the regions, except for the western ones. About 12% of vines were found in steppes (M11) in Kakheti and Kvemo Kartli, 10% in mesophytic and hygromesophytic coniferous and mixed broad-leaved–coniferous forests (D32 and D33) in western and central Georgia, and 6.7% on the vegetation of flood-plains, estuaries and fresh-water polders, and other moist or wet sites (U22) in Kvemo Kartli and Mtskheta-Mtianeti, while less than 5% were found in xerophytic coniferous forests, woodlands, and scrub (K33) in Kakheti.

Based on this geographical analysis it is therefore possible to say that the real distribution of wild vines in Georgia is consistent with the expected natural vegetation of the species.

**Figure 5 plants-14-00232-f005:**
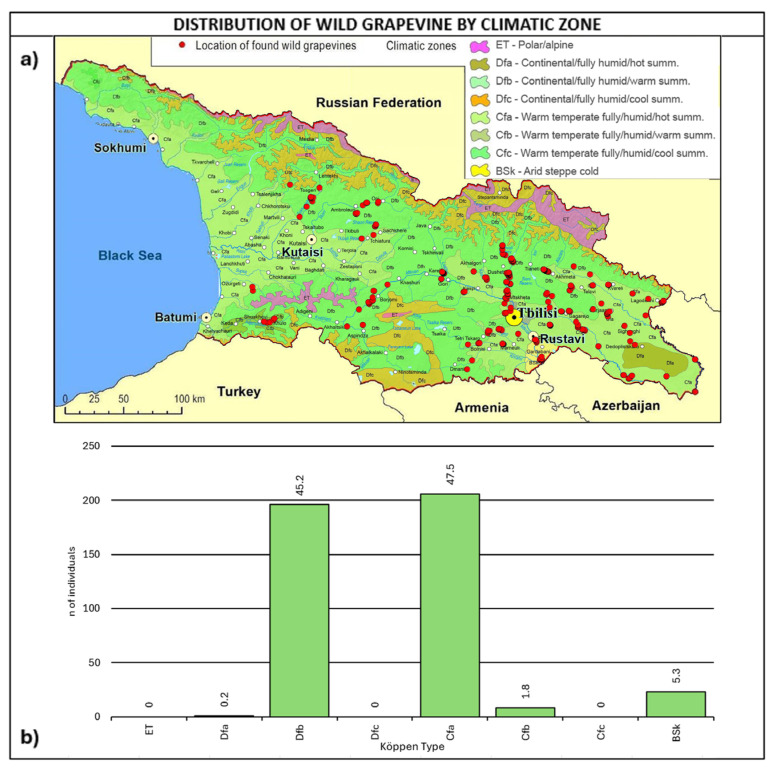
(**a**) Map of wild grapevine sampling sites and Köppen climate types [65,66], (**b**) distribution of the individuals along the climate types. The frequency of distribution (%) is shown above the bars.

**Figure 6 plants-14-00232-f006:**
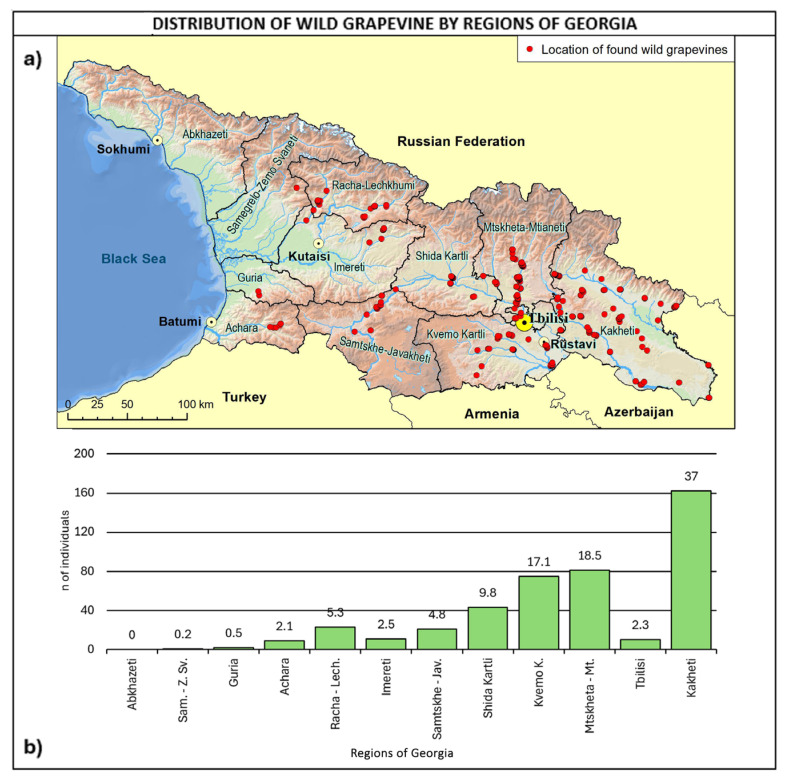
(**a**) Map of wild grapevine sampling sites and administrative regions, (**b**) distribution of the individuals along the regions. The frequency of distribution (%) is shown above the bars.

**Figure 7 plants-14-00232-f007:**
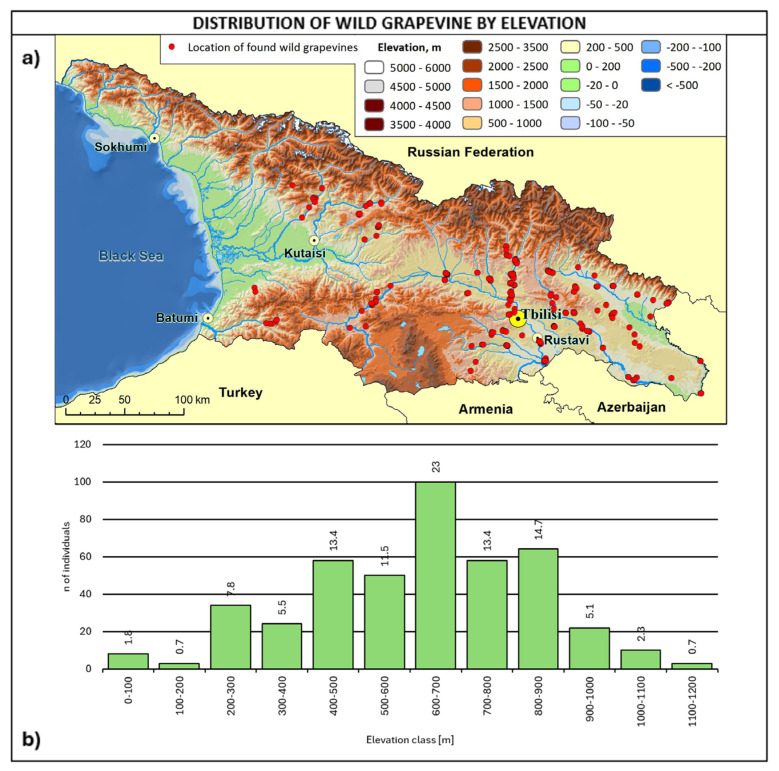
(**a**) Map of wild grapevine sampling sites and elevation, (**b**) distribution of the individuals along the elevation ranges. The frequency of distribution (%) is shown above the bars.

**Figure 8 plants-14-00232-f008:**
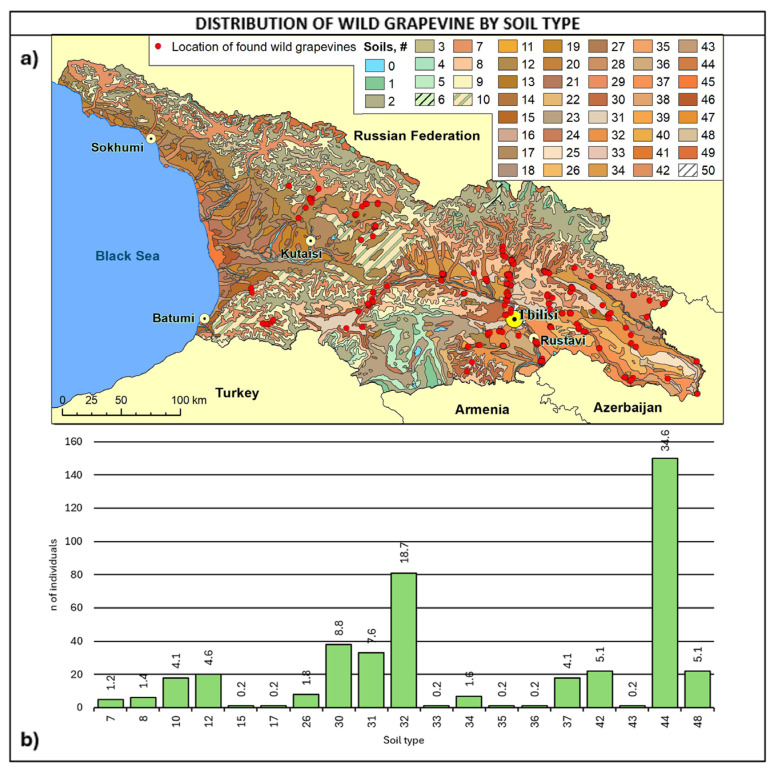
(**a**) Map of wild grapevine sampling sites and soil types, (**b**) distribution of the individuals along the soil types. The frequency of distribution (%) is shown above the bars. Soil classifications [67] provided in Table A4.

**Figure 9 plants-14-00232-f009:**
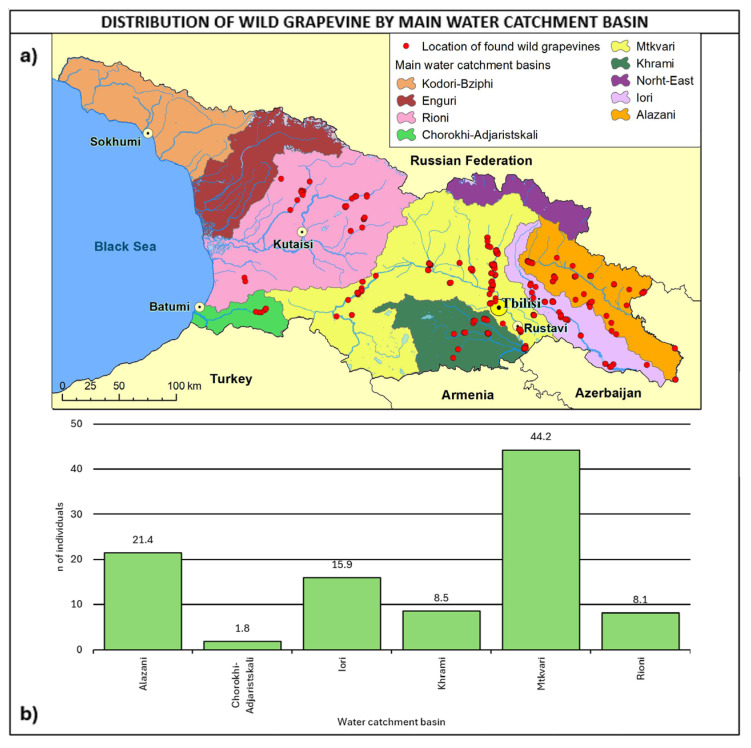
(**a**) Map of wild grapevine sampling sites and main water catchment basins, (**b**) distribution of the individuals along the basins. The frequency of distribution (%) is shown above the bars.

**Figure 10 plants-14-00232-f010:**
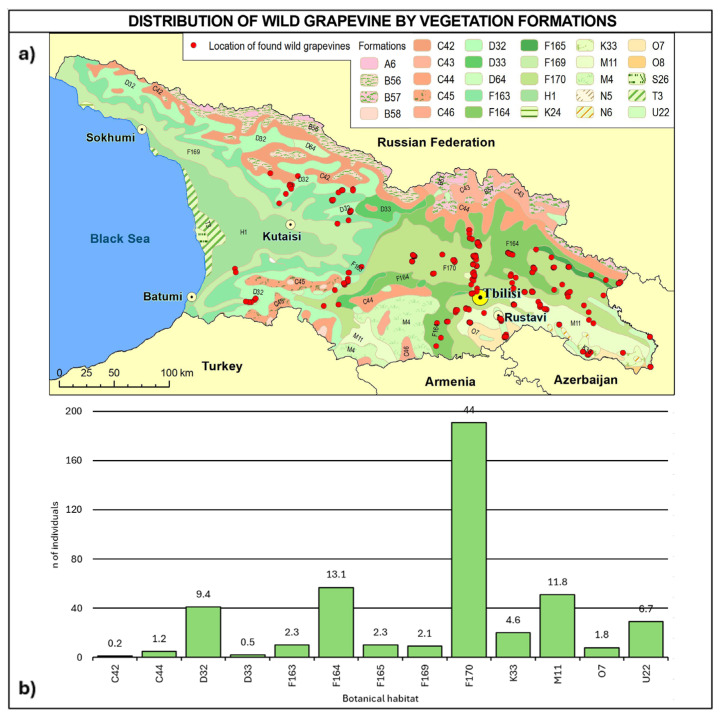
(**a**) Map of wild grapevine sampling sites and vegetation formations, (**b**) distribution of the individuals along the vegetation formations. The frequency of distribution (%) is shown above the bars. Botanical classification [68] is provided in Table A5.

## 3. Discussion

The new surveys in the search for wild grapevine individuals conducted in Georgia during the last two decades confirmed the previous distribution depicted by Ramishvili [55,56], with the only exception being Abkhazeti’s survey, where access was not possible. However, in a qualitative comparison with Ramishvili’s findings, the individuals were rarer and the populations were more scattered than in the past. Most of the sites identified were represented by only one or two plants. Only 17% of the sites host small populations consisting of more than six individuals, up to a maximum of twenty-six, all located in eastern Georgia. Moreover, some of the individuals that were identified in the first surveys have since disappeared, due to the previously discussed causes. The general picture is comparable to the Eurasian situation and active measures are needed to effectively protect and preserve wild grapevine populations.

For approximately a quarter of the surveyed vines, it was possible to determine or infer the sex. Unlike the findings of Anzani et al. [42] for the Italian population, which showed a male-to-female ratio of about 2:1, the Georgian population appeared more balanced, with a ratio slightly below 1 (0.8) and closer to the ratio detected by Ekhvaia and Akhalkatsi [58].

The geographical and ecological analysis confirmed the characteristic wild grapevine habitats, with fully humid warm and hot climates. The preference for humid environments is also evidenced by the prevalence of individual findings close to rivers. Wild grapevines were prevalently found on deep, fertile, and evolved soils, 80% of which are equally divided between alluvial and cinnamonic soils, with a marginal presence on strongly eroded soils. The high presence of wild grapevines close to the rivers could be explained by ecological conditions adverse to the survival of phylloxera [69].

Most vegetation formations are forests and open woodlands, with the southeastern individuals found on steppes (M11).

The altitudinal range is wide, between 0 and 1200, with 80% of individuals normally distributed in the 400–900 m range. It is worth to note that the altitudinal distribution reflects almost the same distribution of cultivated grapevines in Georgia.

Knowledge about the ampelographic characteristics and phenology of wild grapevines is limited, despite significant advancements in understanding their genetic structure over the past twenty years, as documented in numerous scientific publications, including the work of Riaz et al. [51], Dong et al. [53], and, more recently, Röckel et al. [70]. These studies demonstrate a geographic gradient of genetic diversity among populations, which can be attributed to five major eco-geographic regions: South Caucasus, Near East, northern/eastern Europe, central/southern Europe, and western Europe/northern Africa.

The present study therefore represents the first example of the ampelographic characterization of Georgian wild grapevines, highlighting a higher ampelographic variability than previously hypothesized by Zdunić et al. [21]. As for the fruit characteristics, the high prevalence of accessions with berries containing colored pulp, in addition to being unexpected, appears to be of potential interest for the breeding of grapevine varieties intended for red wine production, particularly in environments where anthocyanin accumulation in grape skins is limited.

The mature leaf morphology showed a large polymorphism, even if generally leaves were wedge-shaped or pentagonal, three or five-lobed, with weak goffering and blistering, convex teeth on both sides, open U- or V-shaped sinus angles, low density of prostrate hairs between the main veins, and none or low density of erect hairs on the main veins.

Bunch and berry morphology were rather uniform. Bunches always had very small size and were cylindric in shape, never dense, and never winged. Berries also had very small size and were mainly globular in shape, always blue-black colored, and never aromatic. A surprising trait was the intensity of the red flesh coloration, which mostly ranged from weak to strong, and only rarely was the flesh uncolored.

The 41 accessions analyzed in this study were previously analyzed with specific reference to agronomic traits [71] and resistance to biotic stresses and more specifically to downy and powdery mildew [72,73]. Those works highlighted the oenological value of Georgian wild grapevines, hypothesizing a relevant role along the path of grapevine domestication. With regard to biotic stress, the Georgian genetic pool was revealed to be an important source of resistance that deserves to be studied and consequently exploited.

The phenological analysis revealed significant differences among the accessions but no difference among populations, with only a slight variation in bud-break timing, indicating a high level of synchronicity overall. Flowering timing proved to be the most uniform stage, suggesting minimal environmental pressure on genetic adaptation, despite a first indication of genetic differentiation highlighted by SNP genotyping, as shown in [74].

With reference to morphology, no geographical-based differentiation was found, based on the selected indicators.

## 4. Materials and Methods

### 4.1. Wild Grapevine Survey

Since 2003 several expeditions to survey the Georgian populations of wild grapevines were performed by the research team of the Institute of Horticulture, Viticulture and Oenology of Georgia. The work was carried out in the framework of Biodiversity International Projects [61]. After that, the surveys were supported by international and national projects [37,60,75].

The expeditions included the areas covered by the Ramishvili survey [56] and included the random exploration of previously uncovered areas, mostly river gorges and forest edges, characterized by eco-geographical features typical of wild grapevine habitats.

All the identified plants were geolocated by GPS (WGS 84 Lat/Lon coordinate system) and each grapevine was characterized.

The final dataset included plants retrieved in other surveys performed in the 21st century [58,76,77] to provide a complete picture of the distribution of wild grapevines in the last 20 years.

The surveys conducted during flowering allowed the determination of plant sex into the categories of “male” or “female”. Occasionally, observations were made only after flowering, making the determination of plant sex more uncertain. In such cases, the sex was classified, where possible, as “probable female” when clusters were present or “probable male” when clusters could be observed at the end of flowering without a berry set. Otherwise, the sex was classified as “unknown”.

### 4.2. The Jighaura Collection

The collection (FAO code is GEO038) belongs to the LEPL Scientific-Research Center of Agriculture and is located in the village of Jighaura (latitude 41.92, longitude 44.76, 513 m a.s.l.) in the Mtskheta municipality in the Inner Kartli province and named after the academician S. Cholokashvili. The collection holds more than 1300 accessions of *Vitis vinifera* grape varieties from Georgia and other countries.

The collection of wild grapevines started in 2014 with 70 accessions of wildly growing grapevines in the adult stage [71]. Among these, 41 accessions of wild grapevine originating from 12 different native populations of Georgia (Table 1 and Figure A2) were included in this study. The accessions are composed of three to five plants each, including both male and female plants. Most of the accessions are from the eastern regions of Kartli and Kakheti.

In wintertime, woody cuttings from all the wild grapevines were propagated by grafting onto rootstock Kober 5BB (*Vitis berlandieri* × *Vitis riparia*). The collected canes of each vine were grafted using traditional techniques for winter budding in nurseries. Grafted plantlets were grown in a nursery field during the entire vegetative season to obtain traditional dormant bench grafts for 2013. The collection was planted in the field in the spring of 2014.

In the germplasm collection, vines are trained with a vertical shoot position trellis system with a 2.3 m × 1.3 m spacing and an N–S row orientation. Specifically, the training system is a bilateral Guyot without spurs, left after winter pruning 20–24 buds/vine.

Germplasm collection management ensures good plant development and adequate grape production by regular winter pruning, canopy and soil management, as well as plant protection.

### 4.3. Ampelographical and Phenological Records

Forty-one accessions, representing twenty populations, were selected from the Jighaura germplasm collection (GEO038). Only accessions with fully developed and good vigor vines were chosen for the ampelographic records (Table 1 and Figure A1) by the OIV [63] harmonized descriptors. Ten young shoots, mature leaves, shoots, bunches, and 50–100 berries of each accession were sampled and subjected to ampelographic characterization in 2019–2021 using 52 descriptors from the OIV *Vitis* descriptor list, selected by the COST FA1003 Project [60] (Table A1). The sex of flowers was detected as one of the main ampelographic markers for the identification of true-to-type *Vitis vinifera* subsp. *sylvestris* [78]. The photos of grapevine organs were recorded in the field and in the lab and are included in the ampelographic cards of the collection [71].

The phenological development of accessions was recorded using the BBCH scale [63] as suggested by COST FA1003 Project [76,79]. A homogenous dataset of phenological development for 32 accessions (Figure A2) was obtained for 2019, 2020, and 2021. The analysis focused on the days of occurrence of the phenological stages BBCH 1, 9, 61, 65, 69, 71, 75, 79, 81, 85, and 89. For each phenological stage, the anomaly of the single accession from the average of all the populations was calculated.

Data have been analyzed considering their relationship with district, viticultural zone, longitude, altitude, and sex, focusing on BBCH stages 9, 65, 75 and 85.

### 4.4. The Georgian Wild Grapevine Populations Database

The database of current wild grapevine distribution includes 434 records and is composed of the results of new expeditions, together with the information derived from previous surveys made in the 21st century [58,77,80] to provide a comprehensive picture of current wild grapevine distribution.

Survey sites have been related to the following features of the Georgian territory by means of ArcGis elaboration:Administrative regionElevationMain water catchment basinVegetation formationSoil typeKöppen climate type

The elevation information was derived from the digital terrain model of Georgia, obtained from NASA SRTM 1 arc-second (30 m) [81].

The Köppen climate classification [65,66] for Georgia was produced by the Department of Hydrometeorology of the National Environmental Agency of Georgia.

The soil classification (Table A4) was based on the Soil Map of Georgia (Scale 1:500,000), produced by the Mikheil Sabashvili Institute of Soil Science, Agrochemistry and Melioration and the Institute of Land Organisation [67].

The classification of the vegetation formations (Table A5) was based on the Map of Vegetation Formations of Georgia from the Institute of Botany of the Georgian Academy of Science [68].

## 5. Conclusions

The geographical and ecological survey of Georgian wild grapevine populations revealed that these plants predominantly thrive in humid environments characterized by warm and fully humid climates, often in proximity to internal water bodies. They prefer deep, fertile, and well-developed soils, primarily alluvial and cinnamonic types, with only a minimal presence on heavily eroded soils. The primary natural habitats include forests and open woodlands, though some individuals are also found in the southeastern steppe region. The altitudinal range extends from sea level to 1200 m, with 80% of the grapevines concentrated between 400 and 900 m.

The territorial survey on the distribution of wild grapevines in Georgia highlighted that, at least in the western part of the country, their presence has significantly diminished compared to information dating back to the mid-20th century.

The most concerning aspect, however, is the extreme fragmentation of the individuals, in particular in West Georgia, with the majority (45%) found as single plants, and therefore unable to reproduce due to the dioecious nature of the species. Even in sites where two individuals have been identified (18%), the survival prospects for the species appear extremely limited. Some hope can be placed in other situations, particularly in sites where the number of individuals exceeds six (18%).

As recently confirmed, the South Caucasus is a primary center for the domestication of *Vitis vinifera* [53]. The importance of this germplasm has recently been highlighted both as a source of resistance to powdery and downy mildew fungal diseases [72,73] and for enhancing the oenological potential of grapes intended for the production of red wines for aging, due to their high polyphenol content [71]. Therefore, we believe it is urgent, even for Georgia, to implement specific protection measures for such an important genetic resource [44].

## Figures and Tables

**Figure 1 plants-14-00232-f001:**
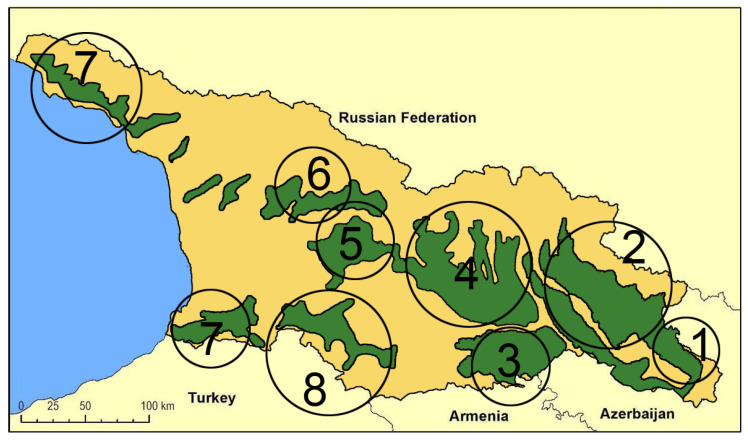
Distribution map of wild *Vitis vinifera* L. populations in Georgia for the second half of the 20th century [57]. (1) Saingilo, (2) Kakheti—the banks of Alazani and Iori rivers, (3) Lower Kartli, (4) Inner Kartli, (5) Upper Imereti, (6), Racha-Lechkhumi, (7) the Black See Regions of Adjara and Abkhazeti, (8) Samtskhe–Javakheti (i.e., Meskheti).

**Figure 2 plants-14-00232-f002:**
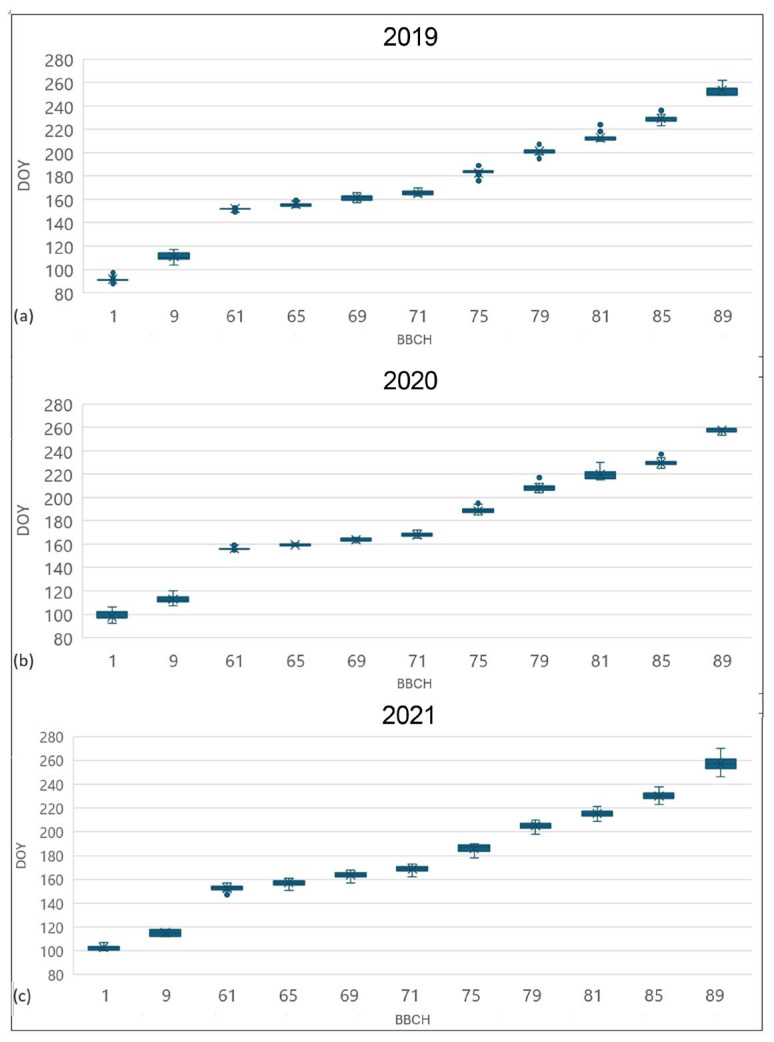
Box and whiskers plot, showing the population phenological course of 2019 (**a**), 2020 (**b**), and 2021 (**c**). For each value, X represents the average, the horizontal line is the median, and the box extends from upper to lower quartile. The whiskers (vertical lines outside the box) represent data variability outside the upper and lower quartiles. Points outside the whisker line represent the outlier data. Legend: 1 = beginning of bud swelling; 9 = bud break; 61 = beginning of flowering; 65 = full flowering; 71 = fruit set; 75 = berries pea-sized; 79 = majority of berries touching; 81 = beginning of ripening; 85 = softening of berries; 89 = berries ripe.

**Figure 3 plants-14-00232-f003:**
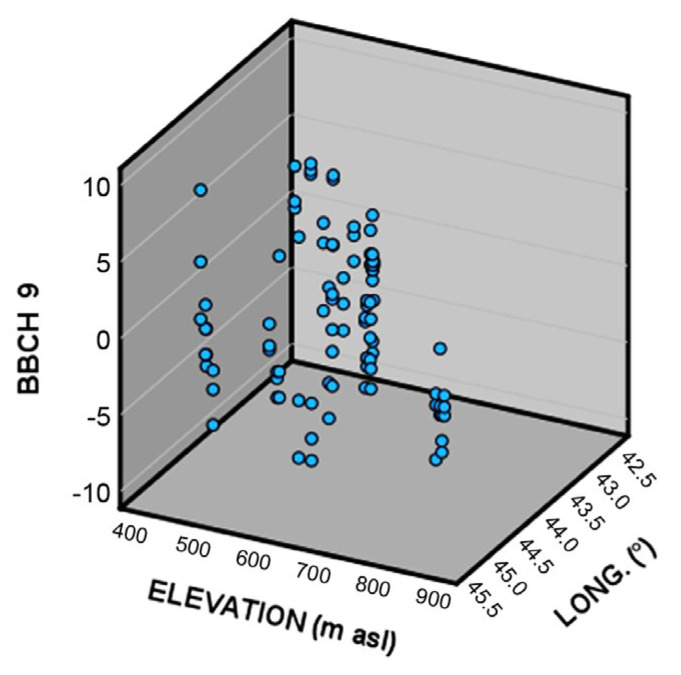
3D scatter plot of the deviation from the population average of the date of occurrence of BBCH 9 (bud break) as a function of elevation and longitude.

**Figure 4 plants-14-00232-f004:**
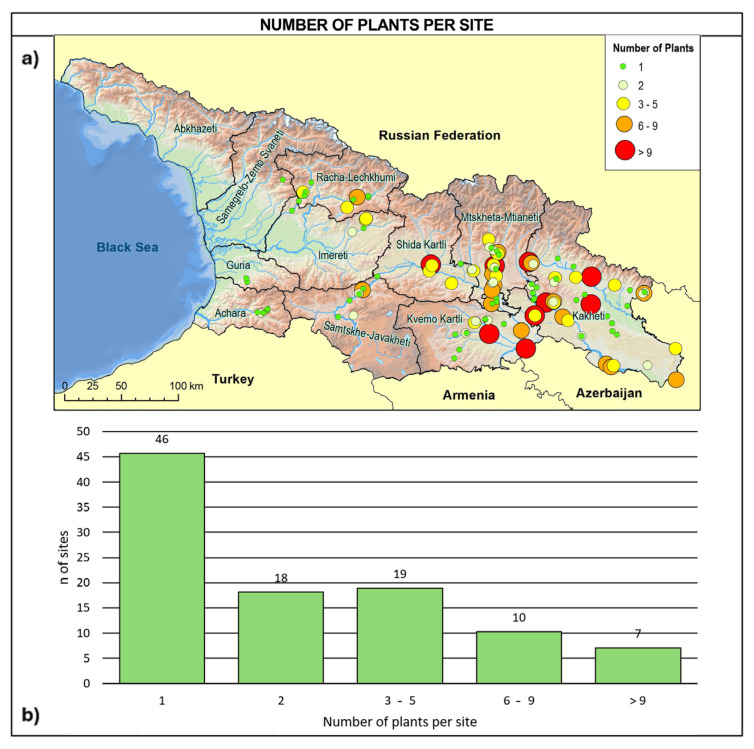
(**a**) Site distribution in relation to the number of wild vines per site and (**b**) frequency distribution of the detected sites in relation to the number of individuals growing in the site.

**Table 1 plants-14-00232-t001:** Main statistics of the OIV ampelographic descriptors selected for the survey [62].

Descriptor	N	Mode	Variance	Min.	Max.
001 Young Shoot: aperture of tip	41	5	0.000	5	5
003 Young shoot: intensity of anthocyanin coloration on prostrate hairs of the shoot tip	41	1	1.605	1	7
004 Young shoot: density of prostrate hairs on the shoot tip	41	5	3.190	1	9
006 Shoot: attitude (before tying)	41	3	0.702	3	5
007 Shoot: color of the dorsal side of internodes	41	2	0.230	2	3
008 Shoot: color of the ventral side of internodes	41	1	0.110	1	2
016 Shoot: number of consecutive tendrils	41	1	0.000	1	1
051 Young leaf: color of upper side of blade (4th leaf)	41	3	1.355	1	4
053 Young leaf: density of prostrate hairs between main veins on lower side of blade (4th leaf)	41	7	3.888	1	9
151 Flower: sexual organs	41	4	2.199	1	4
153 Inflorescence: number of inflorescences per shoot	37	2	0.646	1	5
155 Shoot: fertility of basal buds (buds 1–3)	40	9	3.272	5	9
067 Mature leaf: shape of blade	41	2	0.449	1	4
068 Mature leaf: number of lobes	41	3	0.430	2	5
070 Mature leaf: area of anthocyanin coloration of main veins on upper side of blade	41	3	1.760	1	5
072 Mature leaf: goffering of blade	41	3	2.039	1	7
074 Mature leaf: profile of blade in cross section	41	3	0.811	1	5
075 Mature leaf: blistering of upper side of blade	41	3	2.678	1	7
076 Mature leaf: shape of teeth	41	3	1.105	2	5
079 Mature leaf: degree of opening/overlapping of petiole sinus	41	3	0.498	1	5
080 Mature leaf: shape of base of petiole sinus	41	1–3	0.950	1	3
081-1 Mature leaf: teeth in the petiole sinus	41	1	0.000	1	1
081-2 Mature leaf: petiole sinus base limited by veins	41	1	0.422	1	3
083-2 Mature leaf: teeth in the upper lateral sinuses	41	1	3.901	1	9
084 Mature leaf: density of prostrate hairs between the main veins on lower side of blade	41	3	2.000	1	7
087 Mature leaf: density of erect hairs on main veins on lower side of blade	41	1	1.195	1	5
094 Mature leaf: depth of upper lateral sinuses	41	3	2.805	1	7
103 Woody shoot: main color	41	2	0.220	1	4
202 Bunch: length (peduncle excluded)	22	1	1.394	1	5
203 Bunch: width	22	1	0.970	1	3
204 Bunch: density	22	3	1.706	1	5
206 Bunch: length of peduncle of primary bunch	22	3	0.981	1	5
208 Bunch: shape	22	1	0.000	1	1
209 Bunch: number of wings of the primary bunch	22	2	0.000	2	2
220 Berry: length	22	3	0.970	1	3
221 Berry: width	22	3	1.013	1	3
223 Berry: shape	21	2	0.148	1	3
225 Berry: color of skin	22	6	0.000	6	6
231 Berry: intensity of the anthocyanin coloration of flesh	21	5	3.429	1	7
235 Berry: firmness of flesh	22	1	0.357	1	3
236 Berry: particularity of flavor	22	1	0.000	1	1
241 Berry: formation of seeds	22	3	0.000	3	3
502 Bunch: weight of a single bunch	22	1	0.000	1	1
503 Berry: single berry weight	22	1	0.000	1	1
505 Sugar content of must	21	9	0.648	7	9
506 Total acidity of must	21	5	1.048	5	7
508 Must pH	21	5	1.562	3	7

**Table 2 plants-14-00232-t002:** Population statistics on phenology for 2019, 2020 and 2021.

BBCH	Statistics	2019	2020	2021
9 bud break	Average [DOY]	111.1	112.7	115.4
	Minimum [DOY]	104	107	112
	Maximum [DOY]	117	120	118
	Standard deviation [n days]	3.7	3.3	2.7
65 full flowering	Average [DOY]	155.7	159.3	157.1
	Minimum [DOY]	153	157	151
	Maximum [DOY]	159	162	161
	Standard deviation [n days]	1.3	1.0	2.4
75 berries pea-sized	Average [DOY]	182.5	188.8	185.8
	Minimum [DOY]	176	185	178
	Maximum [DOY]	189	195	190
	Standard deviation [n days]	3.4	2.2	3.5
85 softening of berries	Average [DOY]	229.1	229.4	230.1
	Minimum [DOY]	223	220	223
	Maximum [DOY]	236	237	238
	Standard deviation [n days]	2.4	3.4	4.1

**Table 3 plants-14-00232-t003:** Multiple regression ANOVA and model coefficients of the standardized timing of bud break (BBCH 9 phenological stage) in relation to the longitude and the elevation of the native sites of the wild grapevine accessions.

Model	Sum of Squares		gl	Quadratic Mean	F		Sign.
Regression	91.343		2	45.672	4.824		0.010
Residual	880.532		93	9.468			
Total	971.875		95				
Model	Unstandardized coefficients			Standardized coefficients		t	Sign.
	B	Standard Error		Beta			
(Costant)	61.885	19.986				3.09	0.003
Elevation (m)	0.004	0.003		0.168		1.38	0.168
Longitude (°)	−1.447	0.471		−0.371		−3.07	0.003

## Data Availability

The raw data supporting the conclusions of this article will be made available by the authors on request.

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
