# Peer review of "The Status of Wild Grapevine (Vitis vinifera L. subsp. sylvestris (C.C. Gmel.) Hegi) Populations in Georgia (South Caucasus)"

_plants, 2025, doi:10.3390/plants14020232_

Round 1

Reviewer 1 Report

Comments and Suggestions for Authors

The paper is well written I have the following minors comments;
1. The study should clearly articulate its primary objectives in the introduction to provide a focused direction for the research.
2. 
The methodology section needs more detailed descriptions of the sampling techniques and statistical analyses employed to ensure reproducibility.
3. 
The discussion should delve deeper into how specific environmental factors influence the distribution and characteristics of wild grapevines. These citation should be cited i.e., 1) DOI: 10.11646/phytotaxa.538.3.2 2) 10.3390/agronomy12051078 
4. The literature review could benefit from a broader range of sources, including recent studies that may provide additional context or contrasting viewpoints.
5. The conclusion should summarize key findings more effectively and suggest practical implications or future research directions based on the results. 

Author Response

We thank reviewer 1 for the suggestions and comments. Specific answeres to the reviewers comment can be found in the attached file.

We want to highlight that we also made some other changes as follows:

1 – Title

We changed Southern Caucasus to South Caucasus, which is widely adopted to define the area

2 – Authors

Regarding the authors order, we moved David Maghradze last author and Gabriella De Lorenzis second.

3 – Habitat

We changed “habitat” to “natural vegetation”. We find this definition more appropriate based on the reference [68].

4 - Section “Consistency and ecology of the Georgian wild grapevine populations”

We apologize for the wrong numbers provided in the first draft. Due to the adoption of a wrong file, the section comments were based on 605 data, with repetitions taken into account. The correct number of individuals is 434. The error did not affect in a significant way the analysis proposed in the first draft, but this new version is based on the correct database analysis.

Reviewer 2 Report

Comments and Suggestions for Authors

Manuscript entitled "The status of wild grapevines (Vitis vinifera L. subsp. sylvestris (C.C. Gmel.) Hegi) populations in Georgia (Southern Caucasus)" gives a valuable insight into the current status of valuable preserved populations of wild grapevine in Georgia. 

The manuscript brings manny detailes on the staus of the populations, and their habitat. Most important ampelogaraphic characteritisc of these populations are presented. 

I suggest only several minor modifications to the manuscript to improve his value to the readers: 

- in current version, resolution of  Apendics A (Figura A1 and A2) graphic is quite low and should be impoved. 

- discusssion must be improved and expanded. Explanation of potential value of specific traits, specific high share of flesh coloration etc. must be provided or discussed in more detailed way. Although the paper references the importance of Georgian wild grapevines to global viticulture and domestication, it provides limited comparation with other regions or broader genetic studies to contextualize findings.

- valuable data on characterisation of resistance to biotic  and abiotic stresses (mentioned in the M&M section) are not presented- if such data are available, they would significantly improve the Significance of Content of this paper to the scientific communiy and breeders. 

- the presentation of the results of statistical processing should be more concise, and some of the tables could be simplified (i.e. table 3 and 4) as well as some charts i.e. figure 2 - cholud be easier to do comparation of the years in same chart.

- conclusions should reflect on some aditional important results (especially if resistance to biotic and abiotic stresses was evaluated.). Conclusions (or better also in discusion)  manuscript should provide specific actionable recommendations for preserving or restoring these populations, recognised as endangered. 

Author Response

We thank reviewer 2 for the suggestions and comments. Specific answeres to the reviewers comment can be found in the attached file.

We want to highlight that we also made some other changes as follows:

1 – Title

We changed Southern Caucasus to South Caucasus, which is widely adopted to define the area

2 – Authors

Regarding the authors order, we moved David Maghradze last author and Gabriella De Lorenzis second.

3 – Habitat

We changed “habitat” to “natural vegetation”. We find this definition more appropriate based on the reference [68].

4 - Section “Consistency and ecology of the Georgian wild grapevine populations”

We apologize for the wrong numbers provided in the first draft. Due to the adoption of a wrong file, the section comments were based on 605 data, with repetitions taken into account. The correct number of individuals is 434. The error did not affect in a significant way the analysis proposed in the first draft, but this new version is based on the correct database analysis.

Reviewer 3 Report

Comments and Suggestions for Authors

This work   concerns the study of the wild grapevines in Georgia.

The dispersion of wild grapevines in Georgia as well as the ecological and soil conditions, in which they grow, were reported. Among the total accessions collected in a germplasm collection, some of them were studied for the ampelographical and phenological traits. The overall work give interesting information on the wild grape populations in Georgia. Wild grapevines are an interesting plant material and could be used to genetic improvement of the vine. The overall work was well designed and the results were analyzed precisely and presented satisfactory. The article is easy to read and includes figures and tables that are well – organized according to the obtained results.  

In my opinion, the manuscript can be accepted with minor revision. I have made some minor comments in the text. 

Author Response

We thank reviewer 3 for the comments and suggestions. Specific answeres to the reviewers comment can be found in the attached file.

We want to highlight that we also made some other changes as follows:

1 – Title

We changed Southern Caucasus to South Caucasus, which is widely adopted to define the area

2 – Authors

Regarding the authors order, we moved David Maghradze last author and Gabriella De Lorenzis second.

3 – Habitat

We changed “habitat” to “natural vegetation”. We find this definition more appropriate based on the reference [68].

4 - Section “Consistency and ecology of the Georgian wild grapevine populations”

We apologize for the wrong numbers provided in the first draft. Due to the adoption of a wrong file, the section comments were based on 605 data, with repetitions taken into account. The correct number of individuals is 434. The error did not affect in a significant way the analysis proposed in the first draft, but this new version is based on the correct database analysis.
